# Tuning the Fröhlich exciton-phonon scattering in monolayer MoS$_2$

Bastian Miller[1,2], Jessica Lindlau[2,3], Max Bommert[1], Andre Neumann[2,3], Hisato Yamaguchi[4], Alexander Holleitner[1,2,5], Alexander Högele[2,3,5] & Ursula Wurstbauer[1,2,6]

Charge carriers in semiconducting transition metal dichalcogenides possess a valley degree of freedom that allows for optoelectronic applications based on the momentum of excitons. At elevated temperatures, scattering by phonons limits valley polarization, making a detailed knowledge about strength and nature of the interaction of excitons with phonons essential. In this work, we directly access exciton-phonon coupling in charge tunable single layer MoS$_2$ devices by polarization resolved Raman spectroscopy. We observe a strong defect mediated coupling between the long-range oscillating electric field induced by the longitudinal optical phonon in the dipolar medium and the exciton. This so-called Fröhlich exciton phonon interaction is suppressed by doping. The suppression correlates with a distinct increase of the degree of valley polarization up to 20% even at elevated temperatures of 220 K. Our result demonstrates a promising strategy to increase the degree of valley polarization towards room temperature valleytronic applications.

[1] Walter Schottky Institute and Physics-Department, Technical University of Munich, Am Coulombwall 4a, 85748 Garching, Germany. [2] Nanosystems Initiative Munich (NIM), Schellingstr. 4, 80799 München, Germany. [3] Fakultät für Physik and Center for NanoScience (CeNS), Ludwig-Maximilians-Universität München, Geschwister-Scholl-Platz 1, 80539 München, Germany. [4] Los Alamos National Laboratory (LANL), Los Alamos, NM 87545, USA. [5] Munich Center for Quantum Science and Technology (MCQST), Schellingstr. 4, 80799 München, Germany. [6] Institute of Physics, University of Münster, Wilhelm-Klemm-Str.10, 48149 Münster, Germany. Correspondence and requests for materials should be addressed to U.W. (email: wurstbauer@wsi.tum.de)

A direct band gap[1], remarkable light-matter coupling[2], as well as strong spin-orbit[3] and Coulomb interaction[4] establish two-dimensional (2D) crystals of transition metal dichalcogenides (TMDCs) as an emerging material class for fundamental studies, as well as novel technological concepts. Valley selective optical excitation allows for optoelectronic applications based on the momentum of excitons[5–9]. For both, optical, as well as electronic phenomena, strength and nature of the electron-phonon interaction are crucial. Scattering with optical phonons dominates the mobility in single layer $MoS_2$ at room-temperature[10,11]. Moreover, electron-phonon coupling plays a fundamental role in the dynamics of photo excited electron hole pairs and related excitons that are bound by strong Coulomb interaction. The electron-phonon or exciton-phonon interaction is of great importance regarding fast cooling of photo-excited carriers[12–14], the homogeneous linewidth of excitonic luminescence[15–17], optical absorption spectra[18] and coherence[19]. In addition to lattice imperfections and disorder[20], scattering by phonons is a significant mechanism for valley depolarization and decoherence of excitons, resulting in a breakdown of valley polarization at temperatures above ~100 K[8,21], thus preventing high-temperature valley polarization required for realistic applications. Exciton-phonon interaction can be directly accessed by resonant Raman spectroscopy, where excitons play an important role as real intermediate states[22].

Here, we combine polarization resolved photoluminescence (PL) with resonant and non-resonant Raman spectroscopy to identify Fröhlich exciton-LO phonon interaction as a significant contribution to valley depolarization via direct exchange interaction in single layer $MoS_2$. We use field effect structures with electrolyte gates that enable a tuning of the free electron density $n_e$ by two orders of magnitude to demonstrate electronic control over the Fröhlich exciton-LO phonon scattering rate and its correlation to the degree of circularly polarization of the PL of the A exciton.

## Results

**Polarization resolved photoluminescence on gated $MoS_2$.** Polarization resolved PL measurements and the resulting degree of polarisation (DoP) are summarized in Fig. 1 for a large range of charge carrier densities at elevated temperature of $T = 220$ K for an excitation energy of $E_i = 1.96$ eV. The PL experiments are carried out on a 1L-$MoS_2$ field effect device utilizing an ionic liquid top gate. Figure 1a shows the circularly co-polarized and cross-polarized PL of the A exciton for applied gate voltages of −2 V and 0 V, corresponding to low and high electron densities, respectively. We estimate an increase of the electron density in the order of ~$10^{13}$ cm$^{-2}$ when increasing $V_{gate}$ by 2 V (Supplementary Fig. 1). For increasing $V_{gate}$, we observe a decrease of the PL intensity, consistent to the well-studied bleaching of the electron radiation interaction for high $n_e$ resulting mainly from Coulomb screening[23]. Figure 1b shows the corresponding spectrally resolved degree of polarization calculated as DoP = $(I(\sigma^+)$ $-I(\sigma^-))/(I(\sigma^+)+I(\sigma^-))$ for a series of $V_{gate}$. In Fig. 1c the DoP at the maxima of the PL peak is plotted as function of the gate voltage together with the DoP of the individual contributions decomposed by a line-shape analysis using Gaussian functions for the neutral (A$^0$) and the charged (A$^-$) excitons (c.f. Supplementary Fig. 7). Taking the values obtained from the total PL signal as a lower limit, we observe an increase of the DoP to up to 20% with increasing charge carrier density $n_e$ by applying a gate voltage of $V_{gate} = 1$ V, while for depletion of the 2D system with negative $V_{gate}$, the DoP is vanishing. The values for the A$^0$ and A$^-$ contributions even reach DoP values of ~60% and ~40%, respectively. According to literature, optically induced valley polarization is

robust only for temperatures up to ~100 K[8], what is consistent with the absence of valley polarization in our measurements at $T = 220$ K for low $n_e$. It is known that resonant pumping increases the DoP[7]. In the presented experiment, however, the energy of the A exciton complex gets slightly more off-resonant for increasing $n_e$ (Fig. 1a). Thus, the resonance energy cannot account for an increasing DoP, which we observed on multiple samples. We therefore investigate the exciton-phonon interaction as a possible depolarizing mechanism in dependence of $n_e$ by means of Raman spectroscopy.

**Polarization resolved resonant Raman spectroscopy.** The Raman active optical phonon modes in 1L-$MoS_2$ visible in backscattering configuration are an out-of-plane oscillation $A'_1$ and the in-plane mode $E'$ that is represented by one longitudinal optical (LO) and one transverse optical (TO) phonon branch (Fig. 2a). Phonons interact with electrons via the deformation potential (DP)[24]. Additionally, in polar crystals such as TMDCs, longitudinal optical (LO) phonons induce a macroscopic electric field which can strongly couple to electrons or excitons via the Fröhlich interaction (FI)[25] as sketched in Fig. 2b. In polarization resolved light scattering experiments, the observed intensity is determined by

$$I \propto |\hat{\mathbf{e}}_s \cdot \mathcal{R} \cdot \hat{\mathbf{e}}_i|^2, \tag{1}$$

where $\hat{\mathbf{e}}_i$ and $\hat{\mathbf{e}}_s$ are the electric field vectors of the incident and the scattered light and $\mathcal{R}$ is the tensor of the scattering interaction, which in the case of DP interaction represents the symmetry of the phonon mode. For the $A'_1$ and the $E'$ phonons, the DP Raman

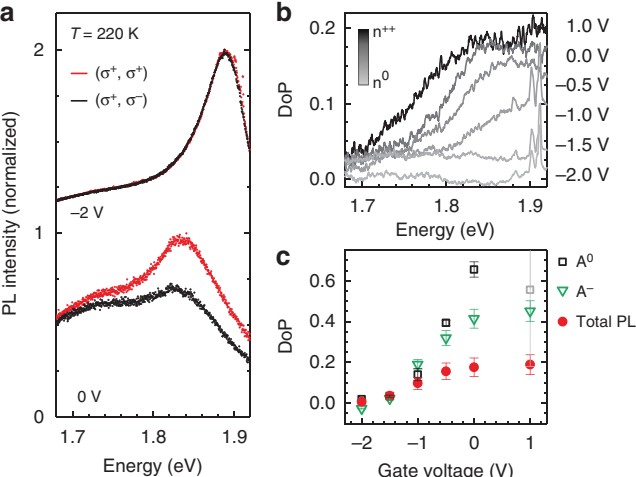

**Fig. 1** Valley polarization in dependence of the electron density. **a** Circularly polarized PL spectra for $\sigma^+$ excitation and $\sigma^+$ (co-polarized) and $\sigma^-$ (cross-polarized) detection measured with $E_i = 1.96$ eV and at $T = 220$ K for two different gate voltages of a 1L-$MoS_2$ device with ionic liquid top gate. Spectra are normalized to the maximum of the respective co-polarized spectrum. The absolute intensity of the spectra for −2 V is a factor of 10 higher than for the spectra taken for 0 V. **b** Spectrally resolved degree of polarization for a series of gate voltages. Negative (positive) gate voltages correspond to electron depletion (accumulation). **c** Degree of polarization as a function of the applied top gate voltage of the total PL signal as shown in **b** evaluated at the PL peak maxima, and of the individual contributions of the neutral (A$^0$) and charged (A$^-$) exciton obtained from peak fits. The error bars denote the standard deviation from the fit approach

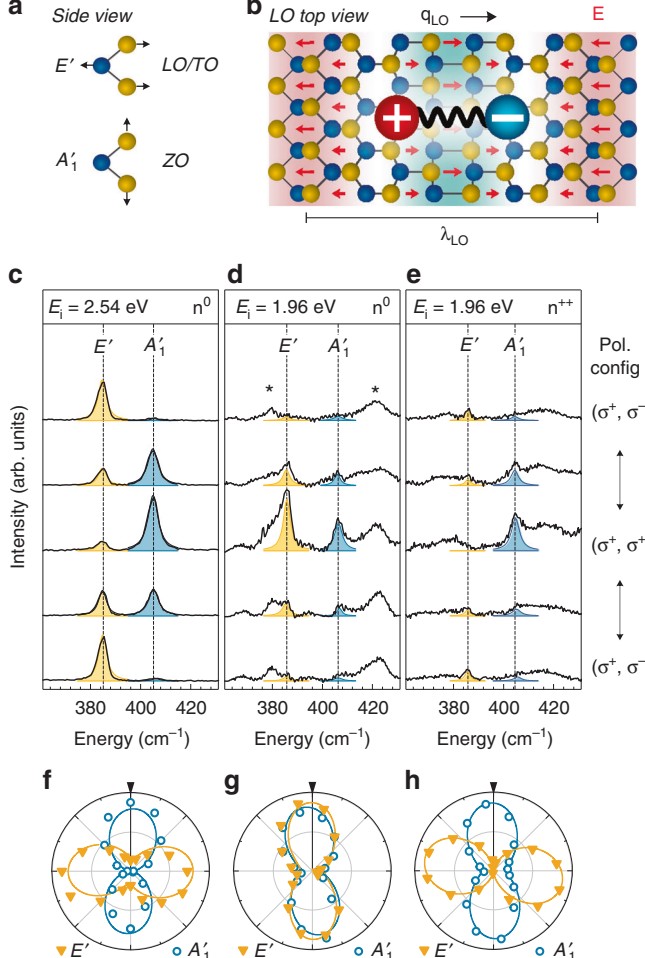

**Fig. 2** Polarization of phonon modes in dependence of the charge carrier density. **a** Raman active optical phonons in MoS₂: the in-plane, polar $E'$ mode and the out-of-plane, homopolar $A'_1$ mode. **b** Illustration of the movement of the atoms for the LO phonon mode. The resulting electric field is indicated with red arrows. The interaction strength between the macroscopic electric field and an exciton depends on the ratio between the exciton radius and the phonon wave vector. **c–e** Polarization resolved Raman spectra for circularly polarized light from a 1L-MoS₂ flake in a field effect device with polymer electrolyte gate at $T = 300$ K. The filled curves are Lorentzian fits to the data. **c** Non-resonant excitation and low charge carrier density $n^0$ ($V_{TG} = -0.5$ V, $V_{BG} = -40$ V). **d** Resonant excitation and low charge carrier density $n^0$. **e** Resonant excitation and high charge carrier density $n^{++}$ ($V_{TG} = 0$ V, $V_{BG} = 0$ V). Asterisks mark additional Raman signatures that are visible under resonant excitation and that are subject to discussion in literature. **f–h** Polar-plots of the normalized amplitude of the fitted peaks shown in the panel above the respective plot versus the rotation of the quarter wave plate. The black arrows mark 0°; 0° and 90° correspond to the $(\sigma^+, \sigma^+)$ and $(\sigma^+, \sigma^-)$ configurations, respectively

tensors are[26]:

$$\boldsymbol{A}_{\mathrm{DP}} = \begin{pmatrix} a & 0 & 0 \\ 0 & a & 0 \\ 0 & 0 & b \end{pmatrix}, \boldsymbol{E}^{\mathrm{LO}}_{\mathrm{DP}} = \begin{pmatrix} 0 & d' & 0 \\ d' & 0 & 0 \\ 0 & 0 & 0 \end{pmatrix}, \boldsymbol{E}^{\mathrm{TO}}_{\mathrm{DP}} = \begin{pmatrix} d & 0 & 0 \\ 0 & -d & 0 \\ 0 & 0 & 0 \end{pmatrix}.$$

(2)

For linearly polarized light, according to Eqs. (1) and (2) the $A'_1$ mode maintains the polarization of the scattered light, whereas light scattered by the $E'$ mode is unpolarized. In the case of circularly polarized incident light, the $A'_1$ mode maintains the

polarization, whereas the $E'$ mode turns circularly right-handed ($\sigma^+$) to circularly left-handed ($\sigma^-$) polarized light[27]. The polarization dependences of the DP tensors are confirmed in non-resonant ($E_i = 2.54$ eV) Raman measurements depicted in Fig. 2c, where the $A'_1$ mode is co-polarized and the $E'$ mode is cross-polarized under circularly polarized excitation. We refer to the configurations $(\hat{e}_i, \hat{e}_s) = (\sigma^+, \sigma^+)$ and $(\sigma^+, \sigma^-)$ as co-polarized and cross-polarized configurations, respectively. The polar plot representation of the normalized mode intensities in Fig. 2f clearly shows the opposite polarization of the $A'_1$ and the $E'$ modes under circular excitation.

In contrast to the DP Raman tensor, the tensor for scattering due to Fröhlich interaction is diagonal[28], hence, the scattering is expected to be co-polarized.

$$\boldsymbol{E}^{\mathrm{LO}}_{\mathrm{FI}} = \begin{pmatrix} c & 0 & 0 \\ 0 & c & 0 \\ 0 & 0 & c \end{pmatrix}.$$

(3)

Consequently, in TMDCs the DP and FI contributions to the LO-phonon scattering are distinguishable by their contrasting polarization selection rules under excitation with circularly polarized light. Indeed, for excitation with $E_i = 1.96$ eV, which is close to the outgoing resonance with the A exciton of MoS₂, we observe a very strong contribution of the $E'$ mode in the co-polarized configuration ($E'_{CO}$) in addition to a rather weak DP related cross-polarized contribution ($E'_{CROSS}$) (Fig. 2d, g). Hence, the $E'$ mode appears to be overall co-polarized. The polarization of the $A'_1$ mode remains unchanged under resonant excitation. Data is taken on a field effect structure (Supplementary Note 1 (sample A)) with a polymer electrolyte top gate at a low electron density $n^0$.

**Doping induced suppression of Fröhlich scattering.** The observed polarization of the $E'$ mode strongly suggests that Fröhlich exciton-LO phonon interaction dominates the Raman scattering over the DP contribution under resonant excitation. Surprisingly, we find a strong suppression of this forbidden Raman scattering for heavily electron doped MoS₂. Figure 2e depicts resonant Raman spectra ($E_i = 1.96$ eV) for an electron density $n^{++}$ that is increased by about two orders of magnitude compared to $n^0$. We estimate the electron density $n_e$ from the energy of the $A'_1$ mode[29]. (Supplementary Fig. 1). For $n^{++}$, the intensities of the DP contributions $A'_{1CO}$ and $E'_{CROSS}$ are in the same order as for $n^0$, but $E'_{CO}$ vanishes completely such that the overall polarization dependence of the $E'$ mode is cross-polarized (Fig. 2h), identical to the non-resonant spectra. In non-resonant Raman measurements, there is no change of the polarization in dependence of $n_e$ (Supplementary Fig. 2). The forbidden Raman signal under resonant excitation and its suppression for large $n_e$ appears in the temperature range from 3 K to 300 K (Supplementary Fig. 3).

**Microscopic origin of the exciton-phonon scattering.** We now turn to the discussion of the microscopic origin of $E'_{CO}$. Strong co-polarized exciton-LO-phonon scattering by FI is known from CdS, GaAs and other semiconductors[30,31], however, due to low exciton binding energies, only at low temperatures. The combined electron-phonon FI for an electron-hole pair cancels out exactly for zero phonon wave vector $q$[32] and only the finite wave vector of the photon makes exciton-phonon Fröhlich scattering allowed in backscattering. Figure 3a shows the dependence of the Fröhlich exciton-LO phonon matrix element on $qa_0$, where $a_0$ is the Bohr radius of the exciton. The interaction is strongest for $qa_0 \approx 2$. In MoS₂, the small exciton Bohr radius in the order of 1 nm[33] results

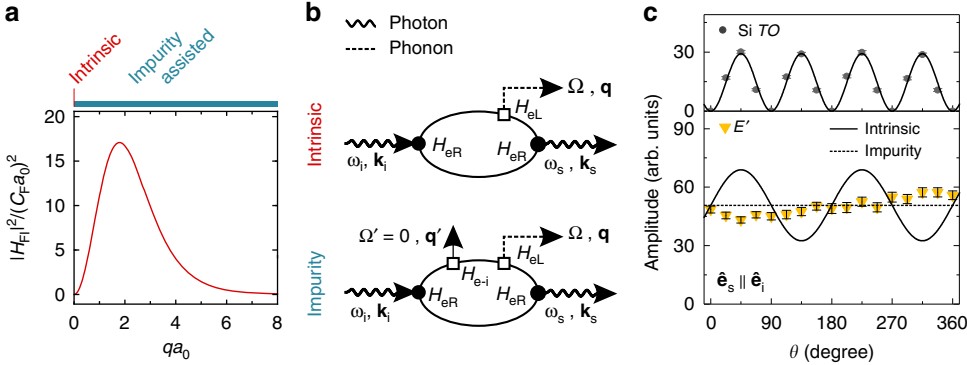

**Fig. 3** Impurity assisted Fröhlich scattering. **a** Plot of the matrix element of the exciton-phonon Fröhlich interaction in dependence of the product of the phonon wave vector $q$ and the exciton radius $a_O$. See Supplementary Note 2 for details. For the intrinsic first-order process $qa_0 \approx 0.02$, while in the impurity assisted process $q$ can take arbitrary values. The $qa_O$ dependence of $|H_{FI}|^2$ is qualitatively independent of $a_O$. **b** Feynman diagrams for the scattering of a photon with frequency $\omega$ and momentum $k$ from initial state i to final state s by emitting a phonon with frequency $\Omega$ and momentum $q$. Upper panel: intrinsic first-order Raman process. Lower panel: second-order process involving elastic scattering with an impurity. $H_{eR}$ denotes the electron-radiation interaction; $H_{eL}$ is the electron lattice interaction, which can be either DP or FI. $H_{e\text{-}i}$ represents the electron-impurity interaction for the elastic scattering with momentum transfer $q'$. We show only one permutation of the interactions. **c** Resonant Raman intensities ($E_i = 1.96$ eV) for linear parallel polarization ($\hat{e}_i = \hat{e}_s$) for one whole rotation of the angle $\theta$ between $\hat{e}_{i,s}$ and the crystal axes. Lower panel: amplitude of the $E'$ mode (yellow triangles). Fitted amplitudes are plotted as scatters. The line plots show the calculated polarization dependences of the intrinsic and the impurity-assisted exciton-LO phonon scattering processes (solid and dashed lines, respectively). Upper panel: amplitude of the TO mode of the silicon substrate used as a reference signal (Scatters: measured data, Line plot: simulated data). Spectra are shown in Supplementary Fig. 4

in $qa_0 = 0.02$ for a first-order Raman process with a photon energy of $E_i = 1.96$ eV, thus, the interaction strength should be weak. However, besides this intrinsic Fröhlich exciton-LO phonon scattering, Gogolin and Rashba[34] proposed a second-order Raman process, involving Fröhlich exciton-LO phonon scattering and a second, elastic scattering process due to electron-impurity interaction, relaxing the momentum-conservation. Figure 3b shows the two Feynman diagrams of the intrinsic and the impurity-assisted Fröhlich exciton-phonon Raman processes. Experimentally, the impurity-assisted second order process can be separated from the intrinsic, first order process due to interference effects as pointed out in ref. [31]. First-order scattering processes via DP or FI have the same initial and final states. Therefore, the tensors of the DP Eq. (2) and FI Eq. (3) sum up before squaring in the calculation of the scattering intensity Eq. (1):

$$I \propto \left| \hat{e}_s \cdot \left( E_{DP}^{LO} + E_{FI}^{LO} \right) \cdot \hat{e}_i \right|^2 \quad (4)$$

In contrast, due to larger possible phonon wave vectors, the final states of the impurity-assisted second-order process are different and the scattering intensities sum up after squaring, prohibiting interference effects:

$$I \propto \left| \hat{e}_s \cdot E_{DP}^{LO} \cdot \hat{e}_i \right|^2 + \left| \hat{e}_s \cdot E_{FI}^{LO} \cdot \hat{e}_i \right|^2 \quad (5)$$

For intrinsic FI scattering, the interference in Eq. (4) leads to a variation of $I$ for different orientations of linearly polarized light with respect to the crystal axes of the sample. Figure 3c shows Raman intensities for parallel polarized incident and scattered light ($\hat{e}_i \parallel \hat{e}_s$) for a whole rotation of $\hat{e}_{i(s)}$ in the plane of the $MoS_2$ crystal (spectra shown in Supplementary Fig. 4). We compare the fitted amplitudes of the $E'$ mode to a calculation of the expected intensities for purely intrinsic or purely impurity assisted FI scattering according to Eqs. (4) and (5). For the calculation, we extract the relative amplitudes of the DP and the FI contributions from measurements with circularly polarized light. As a reference, we show the calculated and measured Raman intensities of the silicon TO mode because the Raman tensor of the Si TO mode implies an intrinsic correlation of the scattering intensity and the

orientation of $\hat{e}_{i(s)}$. From the comparison of experiment and calculations, we conclude that the observed forbidden Raman scattering is consistent to an impurity assisted second-order Fröhlich exciton-LO phonon scattering process that activates scattering with large $q$ phonons. We would like to stress, that for an increase of the exciton radius $a_0$ by e.g., a factor of 10 with increasing electron density[23] and the subsequent increase of $qa_0 = 0.2$, the probability of the intrinsic process is only minor increased (Fig. 3a) and remains small. As $q$ is not fixed in the impurity assisted process, the $qa_0$ dependence of this interaction remains valid. Further, we exclude an externally applied off-plane electric field to be responsible for the activation of the Fröhlich interaction, because we do observe the presence and absence of the co-polarized $E'$ phonon mode in resonance Raman scattering for samples with different intrinsic doping levels without the application of an electric field (Supplementary Figs. 5, 6). This large variation in the intrinsic doping level of exfoliated $MoS_2$ monolayers from sample to sample might explain conflicting reports in literature for as-prepared $MoS_2$ monolayers demonstrating the $E'$ phonon being cross-polarized[27] or co-polarized[35] under resonant excitation.

## Discussion

The impurities involved in the scattering process can be either neutral or charged[31]. Electrostatic doping leads to screening of charged impurities, as well as to a filling of (shallow) potential fluctuation, and hence to a reduction of the electron-impurity scattering cross section. As the change of the Fermi energy in our experiments is limited to ~13 meV, we expect shallow potential fluctuations induced by local strain or dielectric modifications due to interfacial imperfection or by the interaction with (charged) impurities in the substrate to be responsible for the impurity assisted Fröhlich scattering process. Additionally to the screening of impurities, the suppression of the FI scattering with increasing $n_e$ might also result from dielectric screening of the FI because the strength of the FI is inverse proportional to the dielectric constant[25]. We can exclude that a shift and broadening of the excitonic resonance and the well-studied bleaching of the

absorption at the exciton resonance, resulting mainly from Coulomb screening[23], to be responsible for the complete suppression of the FI scattering intensity with increasing $n_e$. The extent to which these effects influence the scattering probability can be estimated from the comparison between the DP and the FI contributions (Supplementary Fig. 5), because the electron-radiation interaction and the resonance condition is equally involved in both scattering mechanisms, as we find in temperature dependent Raman and PL measurements (Supplementary Fig. 8). The much stronger suppression of the FI contribution compared to the DP contributions therefore indicates a suppression of the scattering interaction itself. We conclude that impurity screening and/or dielectric screening of the FI are presumably the most relevant effects to account for a complete suppression of the Fröhlich scattering for high $n_e$. For long wavelength phonons, theoretical models predict a screening of the Fröhlich interaction by electron doping[36,37].

The suppression of Fröhlich scattering with increasing $n_e$ coincides with an increase of the DoP of the PL from the A exciton. Excitonic intervalley scattering under electron-hole exchange interaction is forbidden by symmetry. The long-range exchange interaction between electron and hole of an exciton is an efficient exchange mechanism between s and p excitonic states in different valleys[38], whereas s and p states in the same valley do not mix. The strong long-range electric field induced by the LO phonon can efficiently brake the symmetry. The broken symmetry enables this mixing, resulting in a loss of valley polarization via the long-range exchange interaction.

In summary, we observe an increase of the valley polarization of the A exciton with increasing electron density. In corresponding Raman measurements, we find strong polarization forbidden resonant Raman scattering from the LO phonon, which we can attribute to Fröhlich exciton-LO phonon scattering due to an impurity assisted second-order process. Electron doping suppresses this process entirely. We conclude that a reduction of the exciton-phonon scattering rate can improve the degree of valley polarization even at a temperature of 220 K and above. Our experiments demonstrate the relevance of Fröhlich interaction to optical processes in TMDCs and uncover a promising strategy for simultaneously improving valley polarization properties and the charge carrier mobility particularly at elevated temperatures, as required for realistic (opto-) electronic device applications.

## Methods

**Sample preparation**. Data shown in the manuscript is taken on micro-mechanically exfoliated monolayer $MoS_2$ flakes (bulk crystal supplied by SPI). We use a PDMS stamp to transfer the flakes onto silicon substrates with a 300 nm thick $SiO_2$ layer as dielectric (Siltronic AG). Contacts to the $MoS_2$ flake and for the electrolyte top gate are fabricated by standard optical lithography and e-beam evaporation of 5 nm Ti and 30 nm Au. As an electrolyte top gate, we use either a solid polymer electrolyte consisting of poly-(ethylene oxide) and $CsClO_4$ (ratio 1:0.12) or the ionic liquid Diethyl-methyl-(2-methoxyethyl)-ammonium-bis-(tri-fluormethylsulfonyl)-imid (Sigma Aldrich).

**Raman and photoluminescence spectroscopy**. Raman scattering and photo-luminescence measurements are performed in a free beam optical setup using a He/Ne ion laser or a Kr/Ar ion laser (Melles Griot) for resonant and non-resonant excitation, respectively. The excitation power is 50 μW for all measurements. The light is focused onto the sample with a ×50, NA = 0.42 objective (Mitutoyo) on a spot size of <2 μm. Polarization control is realized by a set of linear polarizers and quarter-wave and half-wave plates (Thorlabs). For details refer to Supplementary Note 1 and Supplementary Fig. 1. Light from the sample is filtered by a suitable steep-edge long-pass filter (Semrock) and analyzed by a single grating spectrometer (Princeton Instruments Acton SP2560) with a nitrogen cooled camera (Princeton Instruments, Acton PyLon BR400). For Raman and PL spectra, we use gratings with 1800 lines per mm and 300 lines per mm, respectively. Temperature control is granted by a flow cryostat (CryoVac).

**Electronic control**. Electronic control over the gate potentials during the optical measurements is realized by a two-channel source measurement unit (Keysight Technologies) for top and back gate. The top gate voltage is ramped at a rate of 1 $mVs^{-1}$ to minimize hysteresis effects. Leakage currents are monitored during all optical measurements to ensure electronic stability of the gate.

## Data availability
The data that support the findings of this study are available from the corresponding author on reasonable request.

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

## Acknowledgements

We gratefully acknowledge financial support by the Deutsche Forschungsgemeinschaft (DFG) via excellence clusters *Nanosystems Initiative Munich* and *e-conversion*, DFG projects WU 637/4-1 and HO 3324/9-1, the European Research Council (ERC) under the ERC Grant Agreements no. 336749 and 772195, the Volkswagen Foundation, the Center for NanoScience (CeNS) and LMUinnovativ.

## Author contributions

B.M., J.L. and A.N. performed the measurements. B.M., M.B, J.L., A.N. and H.Y. prepared the samples. B.M., J.L., A.Hoe and U.W. conceived the experiment. B.M., J.L., A.Hoe, A.Hol. and U.W. analyzed the data. B.M. and U.W. prepared the figures and wrote the manuscript. All authors discussed the results and commented on the manuscript.

## Additional information

**Competing interests:** The authors declare no competing interests.

