## [Peer Review File · Nature Communications]

Reviewers' comments:

Reviewer #1 (Remarks to the Author):

The manuscript presents an interesting study to control the valley polarization of monolayer MoS₂ by tuning the Frohlich exciton--phonon interactions. The revised version has well explained my questions and comments in the last-round review. Therefore, I recommend it to be published.

Reviewer #2 (Remarks to the Author):

The reply sounds reasonable and comprehensive. The revised manuscript satisfies the publish standards for Nature comm. I recommend it for publication at the present form.

Reviewer #3 (Remarks to the Author):

I fully understand that authors' claim that the paper's major novelty is based on the correlated observation of the gate tunable E' collinear Raman peak and the emergence of DoP in PL at the elevated temperature, which they attributed to the tunable Frohlich interaction. I acknowledge their effort to improve the manuscript, but am not yet convinced about their major conclusion.

First, I cannot agree with the author that "we exclude that the suppression of the co-polarized LO phonon scattering with increasing electron doping is due to off-resonance". It is clearly shown in the Fig. 2c that in the completely off-resonance excitation condition, negligible collinear E' peak can be observed while the collinear A1' peak is significant. Though Fig. 2c seems to be contradictory to the new temperature dependence data, which itself is a question worth authors' attention, the hypothesis that the suppression of collinear E' is due to the exciton shift, broadening, or filling cannot be excluded.

Furthermore, I want to challenge authors' new fit on the exciton/trion contribution in Fig S7a. Given no distinct double peak/hump feature is observed in any spectrum, the current fitting for A and A- peaks seems to have very large uncertainty and some counterintuitive conclusion. For example, the lineshape of the trion is almost unchanged from -2 to 0 V, but shows a drastic increase at 1 V. Therefore I will challenge the reliability of the fitted exciton peak energy, strength, and polarization, which is critical to one of the main conclusions that both exciton and trion have finite circular polarization at the elevated temperature.

Most importantly, I stick to my previous conclusion that there is no strong evidence in this manuscript showing that the Frohlich interaction is responsible for the valley depolarization at the temperature above 100 K. As an example of the alternative explanations for the emergence of a finite valley polarization, the ionic doping might significantly shorten the nonradiative lifetime, which plays an important role in determining the DoP as authors also agree.

Reviewers' comments:

Reviewer #1 (Remarks to the Author):

The manuscript presents an interesting study to control the valley polarization of monolayer MoS₂ by tuning the Frohlich exciton--phonon interactions. The revised version has well explained my questions and comments in the last-round review. Therefore, I recommend it to be published.

Reviewer #2 (Remarks to the Author):

The reply sounds reasonable and comprehensive. The revised manuscript satisfies the publish standards for Nature comm. I recommend it for publication at the present form.

Reply:

We would like to thank both reviewers for the supportive and positive answers.

Changes to the manuscript:

Does not apply.

Reviewer #3 (Remarks to the Author):

I fully understand that authors' claim that the paper's major novelty is based on the correlated observation of the gate tunable E' collinear Raman peak and the emergence of DoP in PL at the elevated temperature, which they attributed to the tunable Frohlich interaction. I acknowledge their effort to improve the manuscript, but am not yet convinced about their major conclusion.

Reply:

We would like to thank the reviewer for his positive note and further comments to our manuscript. We are convinced to fully address the three points raised by the referee in our point-by-point answers below:

(A)

First, I cannot agree with the author that "we exclude that the suppression of the co-polarized LO phonon scattering with increasing electron doping is due to off-resonance". It is clearly shown in the Fig. 2c that in the completely off-resonance excitation condition, negligible collinear E' peak can be observed while the collinear A1' peak is significant. Though Fig. 2c seems to be contradictory to the new temperature dependence data, which itself is a question worth authors' attention, the hypothesis that the suppression of collinear E' is due to the exciton shift, broadening, or filling cannot be excluded.

Reply:

We would like to recapitulate the twofold role of the resonance conditions for the observation of the described Fröhlich exciton-LO phonon interaction process in Raman scattering. The first order Raman process can be described by second order perturbation theory and involves a first step with an absorbed photon promoting an electron from the valence band to the conduction band described by the irradiation-electron interaction Hamiltonian. In a second step, the electron (or hole) excites a phonon by an inelastic scattering process described by the electron-phonon interaction Hamiltonian and in a third step the scattered electron and the hole recombine under emission of a photon described again by the irradiation-electron Hamiltonian. This process is described in the paper with help of the corresponding Feynman diagram in figure 3(b).

Resonance Raman means that either the incoming or the outgoing photon has the same energy as a fundamental interband transition of the electronic band structure. Resonance Raman impacts the Raman process twofold: Firstly, the light-matter interaction as well as the second order perturbation is then enhanced. Consequently, the intensity of the Raman process depends on the energy and width of the contributing electronic state with respect to the excitation energy. This contribution is described by the first and third step in the scattering process. Secondly, the intermediate state that interacts with the phonon is a real electronic state with distinct properties that are important for the electron-phonon coupling. This is described by the second step in the Raman process.

We now elaborate the impact of both contributions in our measurements and their interpretation.

The intensity of the Raman signal for both, depolarization contribution (depolarization field induced by phonon modes) and Fröhlich (macroscopic field induced by LO phonon in polar media) contribution depends on the light-matter interaction strength of the incoming and outgoing photons as described above. Since Raman is an immediate process, the light-matter interaction strength can be determined from absorption measurements. For MoS₂ monolayers at room temperature, the used excitation wavelength of 633 nm and 638 nm for the resonance Raman measurements are not perfectly matching the peak position of the A-exciton, but there is still a rather high absorbance denoting a high oscillatory strength of the interband transition. But also for 488 nm used for the non-resonance Raman measurements, the absorbance is similarly large, what explains the rather strong Raman signal observed also for the non-resonance Raman data [e.g. PRB B 90, 205422 (2014)]. Thus, the absolute intensities of Raman scattering alone do not allow for a statement about the resonance to a specific electronic state.

For the observation of the Fröhlich exciton-phonon interaction process in Raman studies it is crucial that the intermediate electronic is a s-like excitonic state. In the relevant energy range we use for resonance Raman measurements, the only state that can be excited is the so-called A-exciton state that has a width of about 40 meV at room temperature. This is because the next higher interband-transition, the so-called B-exciton is at higher energies of above 2.1 eV and can consequently not be excited with the used laser energy of 1.96 eV - even in the highly doped regime. This statement is corroborated by several experimental observations. If the B-exciton would be excited to a measurable amount in the highly doped region, the DoP in PL should vanish with increasing doping [Mak, Nat. Nanotechnol. 7, 494 (2012).], but the opposite is the case as we show e.g. in figure 1. In PL measurements, we do not find evidence that the energy of the A-exciton emission does significantly change with doping.

Moreover, the referee notices correctly, that the 'new temperature dependence data' [Fig. S8] shows a behavior of the E' and A₁' modes, that is different to the behavior displayed in Fig. 2(c). We demonstrate in the temperature dependent data in Fig. S8, that a change of the above described impact of the light-matter interaction strength affects the intensity of the E' and the A₁' peak to the same extent [Fig. S8(e, g)]. At the same time, the constant intensity ratio of the co-polarized A₁' mode and co-polarized E' mode versus

temperature demonstrates that the intermediate state in the Raman process is the 1s-exciton (the A-exciton) in the whole temperature range, even if the excitation energy fits not exactly the peak position of the A exciton. From the above discussion and the observation of a significant co-polarized A_1' peak in Fig. 2(c), we conclude, that for high doping there is a finite light matter interaction and, the A exciton state is excited by the laser. Therefore, the fact, that in Fig. 2(c), the co-polarized E' peak is negligible, while the co-polarized A_1' peak is significant [consistent to Fig. S8(h)], demonstrates, that the suppression of the E' mode in Fig. 2(c) is due to a suppression of the Fröhlich scattering.

In this context, the suppression of the Fröhlich interaction with increasing doping is consistent with recent theoretical predictions by T. Sohler and coworkers [Nano Lett., 17, 3758 (2017) and Phys. Rev. Materials 2, 114010 (2018)].

Changes to the manuscript:

At the discussion on page 9, line 16 the following sentence has been added:

„For long wavelength phonons, theoretical models predict a screening of the Fröhlich interaction by electron doping [Nano Lett., 17, 3758 (2017) and Phys. Rev. Materials 2, 114010 (2018)].”

(B)

Furthermore, I want to challenge authors' new fit on the exciton/trion contribution in Fig S7a. Given no distinct double peak/hump feature is observed in any spectrum, the current fitting for A and A- peaks seems to have very large uncertainty and some counterintuitive conclusion. For example, the lineshape of the trion is almost unchanged from -2 to 0 V, but shows a drastic increase at 1 V. Therefore I will challenge the reliability of the fitted exciton peak energy, strength, and polarization, which is critical to one of the main conclusions that both exciton and trion have finite circular polarization at the elevated temperature.

Reply:

First, we want to note, that the trends of the fit parameters for increasing gate voltage are consistent with existing literature [e.g. Mak et al. Nature Materials volume 12, pages 207–211 (2013)]. Still, we agree, that the fits have a large uncertainty, especially for positive gate voltages. For this reason, we only included DOPs of the unfitted data in the first submission. Since – as correctly stated by the referee – such fits are accompanied by some uncertainties, in the argumentation and description in the main text, of the manuscript, we stick to the DOP values retrieved from the unfitted data. These values give a lower limit for the DOP.

Unlike stated by the referee we do not make the explicit claim in the text that exciton and trion have finite circular polarization at the elevated temperature. However, we admit that the fit to the data allows for such an interpretation. We still believe that it is best to show the DOP of the “raw” data, but also provide the line-shape analysis since both procedures result in the very same interpretation that the doping induced suppression of the Fröhlich exciton LO-phonon interaction coincides with the doping induced increase of DOP of the (non-defect related) PL. With that we provide the whole data set in a fully transparent manner to the broader as well as experienced readership that we are aiming to approach with a publication in Nature Communications.

Changes to the manuscript:

Does not apply.

(C)

Most importantly, I stick to my previous conclusion that there is no strong evidence in this manuscript showing that the Frohlich interaction is responsible for the valley depolarization at the temperature above 100 K. As an example of the alternative explanations for the emergence of a finite valley polarization, the ionic doping might significantly shorten the nonradiative lifetime, which plays an important role in determining the DoP as authors also agree.

Reply:

In our reply to this comment, we closely follow the argumentation on the almost identical question raised by referee #2 point 2 in the previous comments. We would like to note that this referee finds our reply “reasonable and comprehensive”.

Excitonic intervalley scattering under electron-hole exchange interaction is forbidden by the preserved C3 symmetry at the K and K' points. According to M. Glazov et al. [M. Glazov et al Phys. Status solidi b, 252, 2349 (2015), M. Glazov et al. Phys. Rev B. 95, 035311 (2017)], the long-range exchange interaction between an electron and a hole is an efficient exchange mechanism for excitons made of electron hole pairs in different valleys via s-p mixing (s and p states of the exciton). The long range electric field induced by the LO phonon in polar media can efficiently break the C3 symmetry. This broken symmetry fosters the s-p exchange (s-state in one valley and p-state in the other valley) resulting in a loss of valley polarization. Moreover, the field of the exciton can efficiently couple to the macroscopic LO field also resulting in an enlarged long-range exchange interaction what would cause a loss of valley polarization [M. Glazov et al. Phys. Rev B. 95, 035311 (2017)]. In such a scenario, in the low doping regime of our experiments, the strong Fröhlich LO-exciton phonon interaction is therefore responsible for the vanishing of the DoP in photoluminescence experiments consistent with a strong co-polarized E' phonon. In the high-doping regime, the co-polarized E' phonon disappears in resonant Raman indicating a significant reduction of the Fröhlich exciton LO-phonon coupling that is expected to cause also a reduced long-range exchange interaction coupling of the excitons between the different valleys. This picture is consistent with the rather high degree of valley polarization observed for the high doping regime and links the reported Raman results with the photoluminescence spectra. Overall, we explain the increased DoP with increasing doping by a suppression of the long-range exchange interaction acting as a major depolarization channel.

We agree with the referee that the deposition of an electrolyte gate can impact the optical properties by changing the dielectric environment and by non-radiative decay channels by the presence of (charged) impurities. In our study, we are aware of this point and first recognized that the dielectric environment is not crucially changed by the solid electrolyte gate. Even more important, based on measurements on several samples that have been prepared by CVD growth or micromechanical exfoliation with and without solid electrolyte gate and that are intrinsically either in the high-doped or low-doped regime we exclude that a non-radiative decay channels induced by the ions in the electrolyte gate significantly impact the observed suppression of the exciton-phonon coupling in resonance Raman measurements [c.f. S3, S5 and S6] or the valley polarization in the PL measurements [c.f. Fig 1 and S5].

The experimentally observed coincidence of the doping induced breakdown of the Fröhlich exciton LO-phonon interaction and the increased DoP at elevated temperatures is in excellent agreement with established theory that a suppression of electron-hole exchange by reduction of the symmetry breaking

macroscopic LO-phonon field induced by the Fröhlich interaction goes along with an increased DoP in PL measurements.

Changes to the manuscript:

No changes to the manuscript required

Further changes to the manuscript:

The abstract and the introduction of the manuscript have been revised to meet the editorial requirements of nature communications without changing the content of the manuscript.